# Probing the limits of cis-acting gene regulation using a model of allelic imbalance quantitative trait loci

**Cathal Seoighe**[1,2]*, **Seán Connaire**[1], **Mehak Chopra**[1,2]

**1** School of Mathematical and Statistical Sciences, University of Galway, Galway, Ireland, **2** Research Ireland Centre for Research Training in Genomics Data Science, University of Galway, Galway, Ireland

* Cathal.Seoighe@universityofgalway.ie

## Abstract

Imbalance in gene expression between alleles is a hallmark of cis-acting expression quantitative trait loci (eQTLs) and several methods have been developed to exploit allelic imbalance to support the identification of eQTLs. Allelic imbalance is also of scientific and, potentially, clinical interest as it can erode the degree to which the effects of deleterious variants are buffered in a diploid organism and has been reported to be associated with the penetrance of pathological genomic variants. Here, we develop and apply a statistical model that is designed to evaluate whether the genotype of a locus is associated with the degree of allelic imbalance of a gene and refer to such loci as allelic imbalance quantitative trait loci (aiQTLs). An advantage of our approach is that it does not depend on linkage disequilibrium between the aiQTL and the associated gene and is, therefore, suited to the identification of eQTLs that act in cis over very large distances. We applied our model to data from the GTEx consortium and examined the relationship between the distance of an eQTL from the TSS of the associated gene and the evidence that the eQTL acts in cis. Previous studies have used a distance of 1Mb from the target gene as an indication that an eQTL acts in cis; however, our results suggest that the majority of eQTLs at distances more than 500 kb from the TSS of the target gene are likely to act in trans (and thus to affect both gene copies). The model used here is also well suited to comparing the overall extent of allelic imbalance between samples. We show that in some tissues allelic imbalance is correlated with age; however, this correlation may be due to changes in the abundance of immune cell populations with age, as we found strong correlations between sample-level allelic imbalance and the inferred abundance of multiple immune cell types across whole blood samples.

## Introduction

Diploid organisms have two copies of most genes; however, the state and activity of the two copies can be very different. This phenomenon, referred to as allelic imbalance, can be observed in many properties of the gene, including in the chromatin state of the DNA from which the gene is transcribed, in the abundance of mRNA derived from each allele of the

---

**Data availability statement:** Genotype and gene expression data, including allele-specifically mapped read counts, were

obtained from GTEx V8 via dbGaP (https://dbgap.ncbi.nlm.nih.gov; Study Accession: phs000424.v8.p2), following data access committee approval (project 20932). The authors did not receive any special privileges in accessing the data that other researchers would not have (subject to data access committee approval). Requests to access the controlled access data can be made via dbGaP (https://dbgap.ncbi.nlm.nih.gov). The R implementation of our model, including example data, is publicly available on GitHub at https://github.com/cseoighe/aiQTL.

**Funding:** This publication has emanated from research conducted with the financial support of Research Ireland under grant numbers 16/IA/4612 (CS) and 18/CRT/6214 (CS and MC). The funders had no role in study design, data collection and analysis, decision to publish, or preparation of the manuscript.

**Competing interests:** The authors have declared that no competing interests exist.

gene, in mRNA splicing and in post-transcriptional regulation [1]. For some genes the allelic imbalance depends on the parent-of-origin of the allele. This is referred to as genetic imprinting and it results in gene expression exclusively or predominantly from the paternally or maternally derived allele [2]. In other cases, the allele that is expressed in a given cell is random. A large number of genes have been found to be affected at least to some extent by this phenomenon of random mono-allelic expression (RMAE) [3,4]. Allelic imbalance can also have a genetic origin, resulting from *cis*-acting genetic variants that affect the expression level of a gene. We have previously suggested that the term allele-specific expression (ASE) should be used exclusively for allelic imbalance with a genetic origin, because the difference in expression is dependent on the alleles themselves [1].

In general, genetic variants that affect gene expression (termed expression quantitative trait loci or eQTLs) can act in *cis* or in *trans* [5–8]. *Cis*-acting variants are correlated with the expression of a nearby gene on the same chromosome, while *trans*-acting variants may be associated with the expression of an unlinked gene. However, it is possible for an eQTL to be located close to the associated gene (or eGene) but for the variant to affect both copies of the gene [9,10]. Such a variant would typically not act directly along the same physical molecule that contains both the variant and gene. For the purposes of the work described here, it is important to distinguish clearly between eQTLs that are co-located with the eGene (which we refer to as proximal eQTLs) and eQTLs that act directly along the same molecule and therefore affect only that copy of the gene that is located on the same chromosome copy [11]. We use the term *cis*-acting variant exclusively for the latter case. Many previous studies have made the assumption tacitly or explicitly that proximal eQTLs act in *cis* [12,13], typically using a threshold of 1 Mb from the transcription start site (TSS) to identify *cis*-acting variants [12,14]. This assumption is likely to be correct in most cases as evidenced by the fact that proximal eQTLs are strongly associated with allelic imbalance [13]. However, the relationship between the distance separating a proximal eQTL from its eGene and the likelihood that the eQTL acts in *cis* remains underexplored.

Because *cis*-eQTLs typically (though not always [9,10]) result in allelic imbalance in gene expression, ASE has been used to assist in the identification of *cis*-eQTLs [11,15–18]. *Trans*-eQTLs, by contrast, affect gene expression in an allele-independent manner, often by altering the activity or expression of factors that regulate that gene [19]. This tends to result in a similar change in expression for both alleles of the gene [20], and thus ASE is not useful for the identification of *trans*-eQTLs. As a method to detect *cis*-acting genetic variants, allelic-imbalance has the advantage of being applicable even for rare variants, where there is insufficient samples containing the alternative allele to provide power to detect an eQTL from a comparison of the total expression level of the gene across samples [12,21]. However, the observation of statistically significant imbalance between alleles is insufficient on its own to infer that the imbalance has a genetic cause, because other sources of allelic imbalance cannot be ruled out. Beyond their usefulness for inferring eQTLs allelic imbalance is also thought to have implications for the penetrance of disease variants in coding regions and haplotypes containing disease variants show evidence of selection for reduced expression [22].

Several statistical models have been developed that can enable imbalance between alleles of a gene to be taken into account for the inference of *cis*-eQTLs [17,18,23]. Most of these methods require phased data and model both the imbalance in expression between haplotypes of the gene as well as the combined expression of the two haplotypes. This requirement for phased data can be a disadvantage for modeling long-range *cis*-acting variants because phasing accuracy begins to break down at length scales of around one megabase [24]. The method proposed here does not require phased data and, instead, focuses on whether the extent of allelic imbalance in the gene is dependent on the genotype of the putative *cis*-eQTL.

An early study of the relationship between allelic imbalance and the genotype of a putative *cis*-acting eQTL first perfomed statistical tests to categorize samples as displaying allelic expression imbalance or not [25]. A contingency test was then performed on the 2X3 contingency table classifying samples by allelic imbalance and genotype of the putative *cis*-regulatory variant. Because this method does not require haplotype inference, it can be applied to *cis*-regulatory variants that are located far away from the eGene; however, it has the disadvantage that binary classification of samples as imbalanced or not is biased by the number of allele-specifically mapped reads in the sample. A more recent study [16] also classified samples as exhibiting allelic imbalance or not and compared the proportion of samples with allelic imbalance between genotype groups at nearby single nucleotide polymorphisms (SNPs), restricting in this case to the comparison of samples that were either heterozygous or homozygous at the SNP, rather than comparing between three genotype groups. Evidence from the difference in this proportion was then combined with evidence from statistical tests of an association with overall expression level to infer *cis*-eQTLs, with the results used successfully to aid in fine-mapping of causal variants at GWAS loci [16]. Again the binary classification of genes as exhibiting imbalance or not has the potential to limit power and to introduce bias arising from differences in the number of mapped reads between samples. Another method, ASEP [26], uses a finite mixture to model allele-specific read counts, assuming that the more highly expressed haplotype is known and consistent across samples. This also requires the assumption of linkage between the causal genetic variant and the affected gene.

Here, we develop a statistical model, based on the symmetric beta distribution, to identify genetic variants that are associated with the extent of allelic imbalance of a gene and refer to these as allelic imbalance quantitative trait loci (aiQTLs). Our method does not require linkage disequilibrium between the eQTL and variants in the affected gene or that the over-expressed allele of the gene is consistent across individuals. Using simulations, we demonstrate that the model can detect *cis*-acting eQTLs. Applying the model to data from the GTEx consortium [13], we determined the relationship between the distance of an eQTL from its target gene and the likelihood that it acts in *cis*. The model also allowed us to identify examples of eQTLs that act in *cis*, despite being separated by relatively large distances from the eGene. Lastly, we explore differences in the overall extent of allelic imbalance between individuals and between tissues and the sources of these differences. Although eQTLs are often reported as putatively *cis*-acting when they are located less than a megabase from the eGene, our results suggest that the majority of eQTLs between half a megabase and one megabase from the eGene do not act in *cis*. The overall extent of allelic imbalance shows a weak correlation with age in some tissues and substantial differences between tissues. In whole blood data we found that the overall degree of allelic imbalance shows relatively strong correlations with inferred immune cell proportions. Imbalance of immune-related genes and differences in immune cell proportions could explain both the observed differences between tissues as well as the weak correlations with age, as the correlation of allelic imbalance with age was no longer significant in whole blood when we correct for immune cell type proportions.

## Data and methods

### Data

Genotype and gene expression data, including allele-specifically mapped read counts, from GTEx V8 were obtained via dbGaP, following data access committee approval (project 20932). We also obtained the complete, open access, set of *cis*-eQTL eGene pairs from the GTEx Portal.

## Model and implementation

The number of reads that map to a specific haplotype of a gene, given the total number of allele-specifically mapped reads is typically modelled as a beta-binomial random variable. The beta-binomial random variable, which has parameters $\alpha$, $\beta$, can be used to model the number of successes observed in $N$ Bernoulli trials when the probability of success in a given trial is a beta random variable with parameters $\alpha$ and $\beta$. In the application to allele-specific expression the successes are reads derived from the A allele (one of the alleles of the eGene is designated arbitrarily as the A allele). We investigated two approaches to the identification of aiQTLs. In the first, we fitted separate beta-binomial distributions for the number of A-allele reads, depending on the genotype of the putative aiQTL, using the vglm function from the VGAM package [27] in R. We then compared the fit to the data of this model to the fit of a model with shared parameters for the beta-binomial random variables between genotype groups. However, we found that this approach could significantly favour the more general model in cases in which just a single sample in one of the genotype groups has unusually strong allelic imbalance. By contrast, we wished to develop a method that can be used to evaluate the evidence for a difference in the proportions of samples with allelic imbalance between the genotypic groups. To do this, we defined a mixture model with likelihood function given by

$$f(A_i|N_i) = \pi \mathcal{B}(A_i; \alpha = \beta = \alpha_1, N_i) + (1 - \pi)\mathcal{B}(A_i; \alpha = \beta = \alpha_2, N_i)$$

where $A_i$ is the number of reads mapping to the A allele in sample $i$, $N_i$ is the total number of allele-specifically mapped reads in sample $i$, and $\mathcal{B}$ is the probability mass function of a beta-binomial random variable. A feature of our approach is that we do not require the identity of the $A$ allele to be shared across samples. We consider the $A$ allele to be chosen arbitrarily and, therefore, model the $A$ allele counts as a symmetric beta-binomial random variable (i.e. we constrain $\alpha$ and $\beta$ to be the same and, thus, we have a random variable with a single free parameter, which we refer to as $\alpha$). The two components of our mixture model differ in the value of $\alpha$, with parameters $\alpha_1$ and $\alpha_2$ in the first and second components, respectively. In each individual the probability that any given read is derived from the $A$ allele of the eGene is treated as a random sample from a mixture of these two symmetric beta distributions (Fig 1).

To test for evidence of association between a genetic variant and the extent of imbalance (i.e for an aiQTL) we compared the fit to the data of a null model in which $\pi$ (the mixing probability) is the the same across all individuals to a model that allows $\pi$ to depend on the genotype, using the log likelihood ratio test. Although this can be done in different ways, we focused primarily on the case in which there were two values of $\pi$, one for homozygotes and the other for heterozygotes. This form of the model is best suited to potentially causal variants in which we expect to see greater imbalance in heterozygous individuals and similar amounts of imbalance in the two homozygous groups. However, this can also pick up variants that are in LD with a causal variant. Given allele-specifically mapped read counts for a set, $S_{hom}$, of individuals homozygous at the eQTL and a set, $S_{het}$, of individuals who are heterozygous at the eQTL, the full log likelihood is given by

$$\sum_{i \in S_{hom}} \log \left( \pi_{hom}\mathcal{B}(A_i; \alpha = \beta = \alpha_1, N_i) + (1 - \pi_{hom})\mathcal{B}(A_i; \alpha = \beta = \alpha_2, N_i) \right)$$
$$+ \sum_{i \in S_{het}} \log \left( \pi_{het}\mathcal{B}(A_i; \alpha = \beta = \alpha_1, N_i) + (1 - \pi_{het})\mathcal{B}(A_i; \alpha = \beta = \alpha_2, N_i) \right)$$

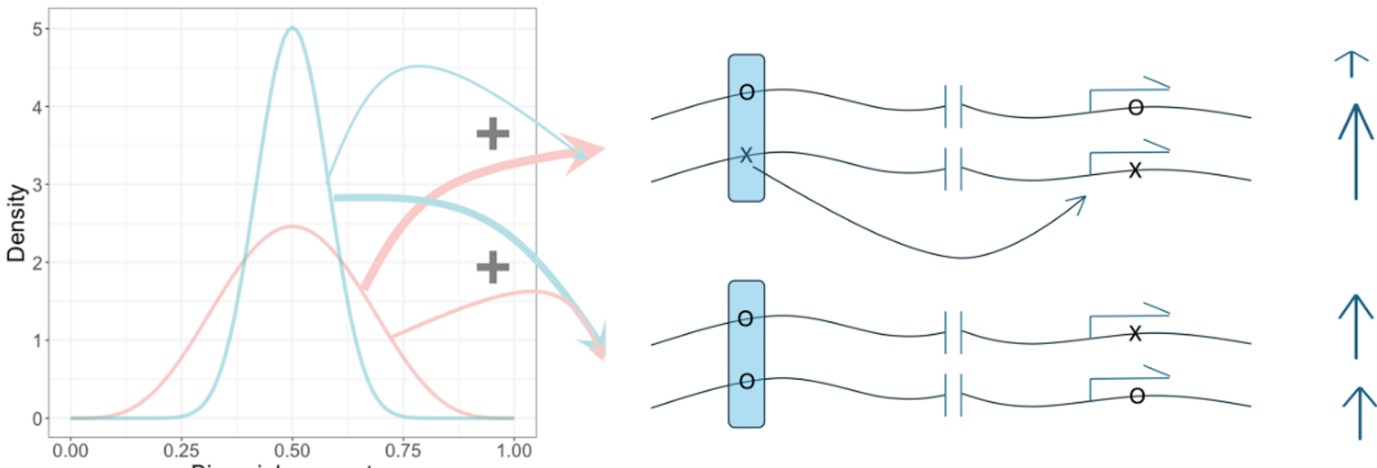

**Fig 1. Graphical illustration of the mixture model.** Here a putative aiQTL (in the shaded box) is being evaluated. The two alleles of the putative aiQTL and of a second variant within the transcribed region (indicated by arrows arising from the transcription start site) are arbitrarily labelled x and o. Allele x of the aiQTL increases the expression of the eGene allele on the same chromosome, resulting in ASE (the expression of each allele is indicated by the heights of the vertical arrows). The allelic imbalance of the gene is modelled using a symmetric beta binomial mixture. The proportion of reads from the x allele of the transcribed variant is sampled from a weighted mixture of the two beta distributions shown on the left of the figure. To evaluate whether the variant of interest is an aiQTL we allow the weights (indicated by the thickness of the arrows emanating from the beta distributions) to depend on whether the putative aiQTL is homozygous or heterozygous. This model is suited to detecting an aiQTL even in the absence of any linkage disequilibrium between the variant and the gene (indicated by the break in the chromosome) and even when there are other genetic variants that also contribute to imbalance between alleles, such that not all of the allelic imbalance is due to the putative aiQTL.

where $\pi_{hom}$ and $\pi_{het}$ are the weights of component 1 of the mixture model in individuals homozygous and heterozygous at the eQTL, respectively. In the null model $\pi_{hom} = \pi_{het}$ and in the alternative model $\pi_{hom}$ and $\pi_{het}$ are independent parameters.

We estimated the parameters of the mixture model by maximizing the likelihood function using the Nelder-Mead method, implemented in the optim function in R. To avoid identifying local minima, we tried multiple starting values of the parameters, including initializing the optimization of the alternative model at optimal values found for the null model and initializing the optimization of the null model at optimal values found for the alternative model (in the latter case we used a weighted average of the $\pi$ parameters from the alternative model to derive the single $\pi$ parameter at which to initialize the optimization of the null model). The R implementation of our model, including example data, is publicly available on GitHub at https://github.com/cseoighe/aiQTL.

## Simulations

Two different approaches were adopted for simulation. In the first we considered two haplotypes with different mean expression levels and treated the number of reads mapped to each haplotype of the gene as independent samples of a negative binomial random variable. This simulation is unrelated to the model that we used to detect aiQTLs and, thus, provides an opportunity to test the performance of the model when the data do not conform to the structure of the model. The characteristics of these simulated data depend on the size parameter of the negative binomial. For a fixed value of the mean, lower values of this parameter result in highly over-dispersed data, whereas in the upper limit of the size parameter the negative binomial becomes identical to the Poisson random variable. Therefore, we simulated over a range of values of the size parameter from 5 (corresponding to highly over-dispersed) to 100

(modest overdispersion). We also simulated different values of the effect size defined, for this simulation, as the ratio of the mean expression of the highly expressed to lowly expressed haplotypes (i.e. the expression fold-change, which was simulated with values of 1.2, 1.3 and 1.5). To test the type-I error rate we performed 1,000 simulations with fold-change of 1 (i.e. no difference in the mean expression of the two haplotypes).

A disadvantage of the above simulation is that it does not take into account that the two alleles are in the same individual resulting in shared *trans*-acting factors and sample characteristics and, therefore, non-independence of the expression levels of the two alleles. This is taken into account by our aiQTL model, which is conditioned on the total number of allele-specifically mapped reads in the sample. In the second simulation type, we assume that the total number of allele-specifically mapped reads is a negative binomial random variable and the number of these reads that map to one of the haplotypes is then a beta-binomial random variable with size parameter given by the total number of allele-specifically mapped reads. We constrained the alpha and beta parameters to be the same (i.e. used a symmetric beta distribution) but the more highly expressed allele was determined at random (to correspond to the absence of LD between the aiQTL and target gene and, correspondingly, no consistency in the over/under-represented allele). To assess the type-I error rate, we carried out simulations in which the single free parameter of this constrained beta-binomial distribution was shared across all samples. For the power simulations, we used a beta-binomial distribution with alpha=beta=100 for the homozygous samples and a separate value of alpha/beta (depending on the simulation) for the beta-binomial random distribution corresponding to the heterozygous samples.

As a further null simulation, we simulated a *trans*-eQTL to ensure that our method would not incorrectly identify an aiQTL in the case of a variant that acts in *trans*. For this simulation, we again used two negative binomial random variables for the total expression level of the gene. The mean of these two random variables depended on the genotype at the regulatory SNP, with the alternative allele having expression that was higher by a factor of 1.5 than the reference allele. As in the previous simulations, we simulated 670 individuals. For these simulations the minor allele frequency of the regulatory SNP was fixed at 0.1.

### Relationship between distance to TSS and probability of allelic imbalance for co-localized eQTLs

We fitted logistic regression models to the probability of a proximal eQTL being detected as an aiQTL as a function of the distance of the eQTL from the TSS. In these models we included the number of samples with at least two allele-specifically mapped reads, the median number of allele-specifically mapped reads among the samples with at least two such reads and the minor allele frequency as predictor variables. Typically, there were multiple proximal eQTLs per gene, resulting in non-independent observations. In such cases, we selected the eQTL with the strongest association among all eQTLs associated with a given gene. We also included the log-transformed p-value for the eQTL association as a covariate in the logistic regression models. The predicted probabilities of a significant aiQTL shown in the Results section correspond to 500 samples, a median of 30 allele-specifically mapped reads per sample and an eQTL with a P-value of $1 \times 10^{-20}$.

### aiQTLs across tissues

The mixture model we developed can be used to assess variation in the imbalance of a gene across tissues and to compare the effect of a given aiQTL across tissues. To compare the

extent of imbalance between tissues we estimated the $\alpha$ parameter separately for each sample and compared the distribution of estimated $\alpha$ values between tissues. We applied gene expression deconvolution to the whole blood samples from GTEx to assess the impact of variation in blood cell type proportions on the extent of allelic imbalance. For this we used CIBERSORTx [28] to estimate cell type proportions from the whole-blood gene expression data of 755 samples. The Ensembl gene IDs were first converted to HGNC IDs using the biomaRt [29,30] and Ensembldb [31] R packages, followed by removal of redundant genes. We used the LM22 leucocyte gene signature matrix [32], corresponding to 547 genes and 22 human hematopoietic cells, as a reference and applied CIBERSORTx using B-mode batch correction with 100 permutations.

## Results

### Model

We identify aiQTLs using a mixture model, consisting of symmetric beta-binomial random variables. The input for the model consists of pairs of counts for each gene in each sample, corresponding to the number of reads that have been mapped to each allele of the gene. Importantly, the two alleles are labelled arbitrarily and the identity of the allele does not need to be the same across each sample. The model is described in detail in the Methods. Briefly, the likelihood of the allele-specific counts observed for a given gene in a given sample is derived from a weighted sum of two beta-binomial random variables, with distinct parameters (Fig 1). We use a symmetric beta-binomial random variable, which is a binomial random variable for which the $p$ parameter is itself a symmetric beta random variable. The symmetric beta random variable has a single parameter, which we refer to here as $\alpha$. Its mean is always 0.5 and variance is inversely proportional to $\alpha$. To test whether a single nucleotide polymorphism is an aiQTL for the gene, we compare the fit of a null model in which the mixture weights are the same for all individuals to the fit of a model in which the mixture weights of the two beta-binomial components are estimated separately for individuals who are heterozygous at the candidate aiQTL and individuals who are either homozygous for the reference or alternative alleles at the locus. The null model is nested in the alternative model and their fits can, therefore, be compared using the likelihood ratio test.

### Results of simulations

Three different null simulations (described in detail in Data and Methods) were performed and in all cases the distribution of the test statistic closely matched the expected null distribution (Fig 2A). These null simulations showed that the model does not detect an aiQTL at loci that are either not associated with the expression of the gene (i.e. not an eQTL) or are eQTLs but do not act in *cis*. The simulations also allowed us to evaluate factors that influence the power to detect an aiQTL. In one type of aiQTL simulation the overall gene expression was a negative binomial random variable with the number of reads from one allele a symmetric beta-binomial random variable with higher variance (i.e. lower value of the $\alpha$ parameter) in heterozygotes than in homozygotes. The power to detect this effect, depended on the frequency of the minor allele and on the magnitude of the difference in the $\alpha$ parameter between the genotypes (Fig 2B,C). In the second set of aiQTL simulations, the expression levels of the two alleles of a gene were modelled as independent negative binomial random variables with the mean of one of the alleles increased by a constant factor (the fold-change illustrated in Fig 2D). This simulation was distinct from the model we developed to identify aiQTLs and more challenging for our method because the over-dispersion which is characteristic of the

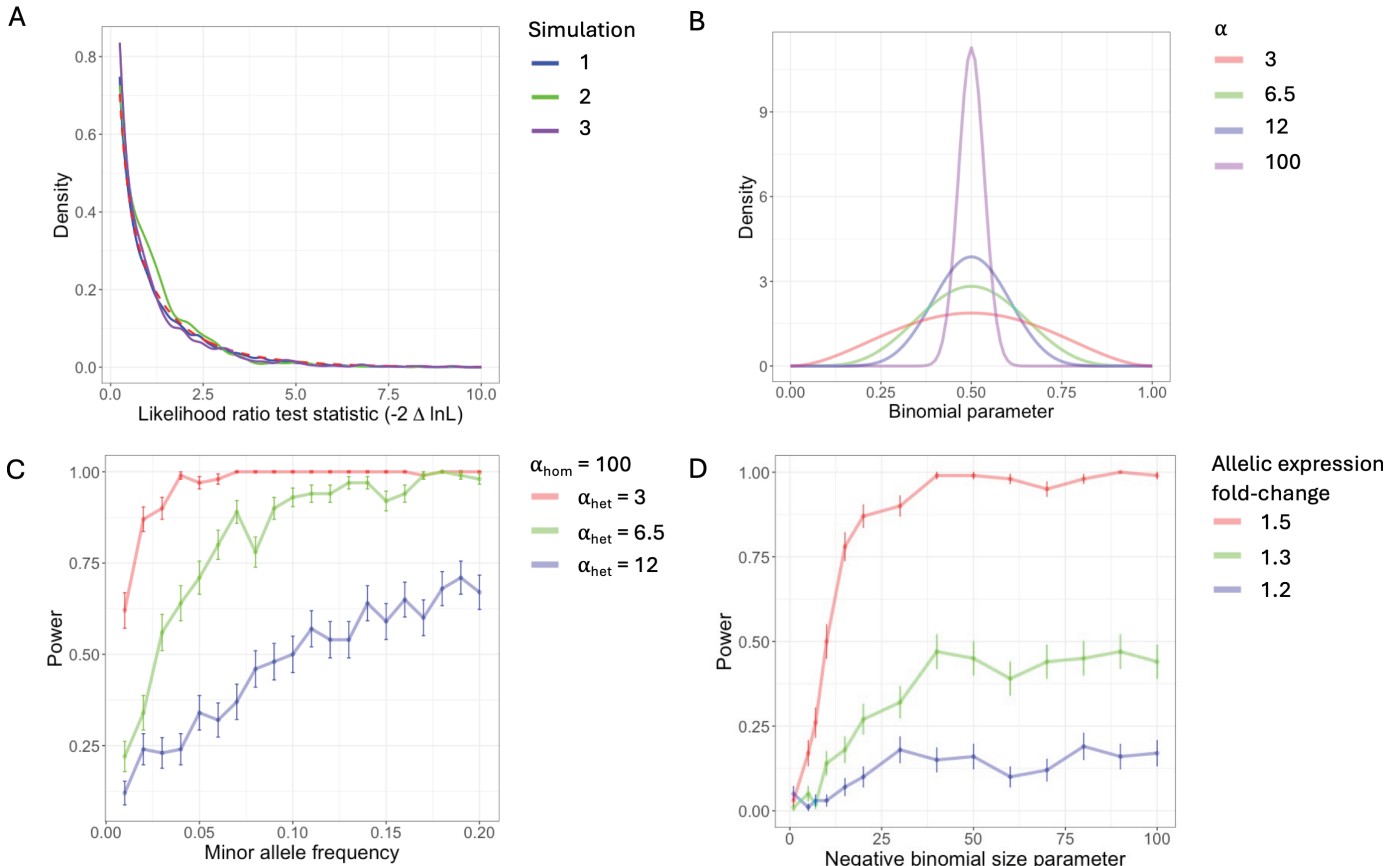

**Fig 2. Simulation results.** A) Null simulations. The results of the three different types of null simulations (described in Data and Methods) are compared to the expected null distribution (chi-squared distribution with one degree of freedom, shown in red dashed line). The simulations correspond to (1) two independent negative binomial distributions for the expression from the two alleles (blue), (2) one negative binomial distribution for the overall expression level, combined with a symmetric beta distribution for the proportion of reads from the reference allele (green) and (3) a simulation corresponding to a *trans*-eQTL (purple). B) Symmetric beta distributions that were used to simulate weak (blue), moderate (green) and strong (red) ASE. In these simulations, the proportion of reads from the reference allele in individuals homozygous at the simulated aiQTL is sampled from the purple beta distribution, while the proportion of reads from the reference allele in individuals heterozygous at the aiQTL is sampled from the blue, green or red beta distribution (depending on the strength of the aiQTL effect being simulated). C) Power as a function of minor allele frequency for the simulations of a single negative binomial (for the total read count) and symmetric beta distributions for the proportions of reads from the reference allele. The red, green and blue lines correspond to the symmetric beta distributions of the same colours shown in panel B. D) Power as a function of the negative binomial size parameter for the simulations of two independent negative-binomial distributions with small (blue), moderate (green) and large (red) expression fold-change between the two alleles. The expression fold-changes are shown in the legend.

negative binomial random variable could result in very different expression values for the two alleles of a gene that was unrelated to the genotype of the candidate aiQTL. This simulation is also likely to be more challenging than the real data in the sense that the two alleles of a gene in a diploid share *trans*-acting factors, such that the variance in expression of the gene across samples is likely to be greater than the variance between alleles. Nonetheless, we found that when the size parameter of the negative binomial random variable was sufficiently large (corresponding to modest over-dispersion) the model had good power to detect aiQTLs with moderate effect sizes (Fig 2D).

**Comparison to other methods.** Some previous studies have tested for association between allelic imbalance at an eGene and the genotype of an eQTL [21,25]. One approach is to perform a statistical test (such as the binomial test) to infer allelic imbalance at the eGene and then to compare the proportion of individuals showing significant imbalance at the gene

between groups defined by the genotype of the eQTL [25]. A significant association could be used to infer that the eQTL is an aiQTL (i.e. acts in *cis*). We refer to this approach as the binomial method (for consistency with our method, we compared the imbalance proportion between heterozygotes and homozygotes at the putative aiQTL). An alternative is to use fixed thresholds on the allelic proportions and again make a binary designation of individuals as ASE if the proportion of the reference allele lies outside this threshold. This approach was used to define allelic imbalance in a previous study that used it to assist in fine-mapping regulatory variants [16]. We adopted this definition of of allelic imbalance to define an aiQTL test and refer to this as the threshold method. Following [16], we designate individuals with a reference proportion less than 0.35 or greater than 0.65 as ASE. Lastly, in one study [21] the absolute departure of the allelic proportions from equality at the eGene was compared between heterozygotes and homozygotes of the eQTL using the Wilcoxon rank sum test. We refer to this as the wilcox method. The model-based method introduced here performed better than the other methods in simulations (S1 Fig). More importantly, our simulations demonstrate that all of the other methods can mistake *trans*-acting eQTLs for *cis*-acting eQTLs (S1 Fig). In all cases the ASE signal is not independent of the expression level of the gene and this results in false-positive inferences that variants act in *cis* (S1 Fig). We also compared the results of the different approaches to defining an aiQTL described above on real data from GTEx, using Whole Blood as an example tissue (S2 Fig). Although many of the aiQTLs were identified by all methods, there were also substantial differences between methods, with the wilcox method appearing to be the outlier in many cases. The wilcox method also identified the largest number of putative aiQTLs, but as demonstrated in S1 Fig, the method is prone to generating false positive aiQTLs.

## The probability that a co-located eQTl acts in *cis* as a function of distance from TSS

We investigated the factors that influence whether an eQTL located close to the corresponding eGene (defined as being within 1Mbp of the TSS and referred to here as a proximal eQTL) shows evidence of being an aiQTL using our model. Here, we describe the results obtained for GTEx whole blood samples as an example tissue. Using a logistic regression model, we found that several factors had a highly significant influence on whether a proximal eQTL was detected as an aiQTL; namely, the distance of the eQTL from the TSS ($P = 1 \times 10^{-38}$), the number of samples with more than one allele-specifically mapped read ($P = 1 \times 10^{-88}$), the log P value of the eQTL ($P = 3 \times 10^{-174}$), the minor allele frequency of the eQTL ($P = 2 \times 10^{-44}$) and the median number of allele-specifically mapped reads (among samples with more than one allele-specifically mapped read; $P = 0.008$). These include factors that are likely to relate to the power to detect an aiQTL (e.g. the number of individuals with more than one allele-specifically mapped read) as well as one variable (distance to the TSS) that is likely to relate to whether or not a proximal eQTL acts in *cis*. The density of eQTLs and their statistical significance falls off dramatically as a function of the distance to the TSS (Fig 3A). More importantly for our purposes, the probability that an eQTL was detected as an aiQTL was also a strong function of the distance from the TSS (Fig 3B). The probability of detection of an aiQTL for an eQTL with the parameters given in the figure legend reduces from 0.58 at the TSS to 0.16 at a distance of 500 kb upstream of the TSS. If we assume that all eQTLs at the TSS act in *cis* (an upper bound), this would imply that for these parameter values our power to detect an aiQTL is 0.58, further implying that approximately 0.28 (0.16/0.58) of the eQTLs at a distance of 500 kb act in *cis*. By 1 Mb from the TSS the predicted probability of detecting an aiQTL no longer exceeds the significance level of the aiQTL tests. The figure also suggests a greater

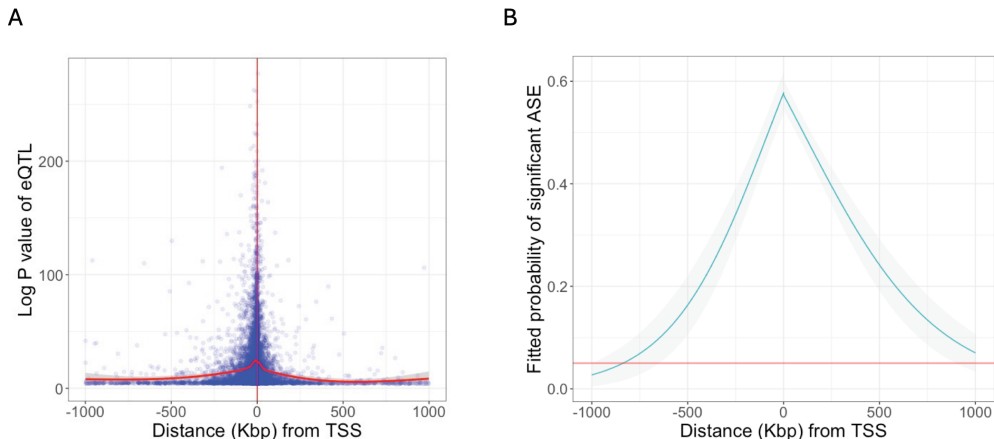

**Fig 3. eQTL status as a function of distance from transcription start site (TSS).** A) Log P-value of GTEx whole blood *cis*-eQTLs as a function of distance from TSS. B) Predicted probability that an aiQTL will be detected for an eQTL as a function of distance from the TSS. Predictions are derived from a logistic regression model, assuming 500 samples with allele-specifically mapped reads, a median of 30 allele-specifically mapped reads per sample and an eQTL with a P-value of $1 \times 10^{-20}$ and minor allele frequency of 0.05.

probability of variants downstream of than upstream acting in *cis* over long distances. This could be because, for some long-range interactions, distance from the gene may be more relevant than distance to the TSS; indeed, variants within the 3′ region may already be quite far from the TSS for some very long genes. Overall, our results suggest that it is unsafe to infer that eQTLs located hundreds of kilobases away from their eGene act in *cis* and that many of the associations detected at these distances are likely to be indirect effects. This is consistent with what has been reported previously [12] using a method that requires phased data and was applied to data inferred using population phasing, accounting for phasing error.

## Examples of potential long-range *cis*-acting eQTLs

Although the majority of eQTLs that are far from the eGene may be acting in *trans*, examples of enhancers that act over large distances have been reported [33–36] and we could find many examples of aiQTLs that were far from the transcription start site (TSS) of the eGene. An example of such a long-range aiQTL is provided in the supplementary information (S3 Fig). In this case the aiQTL is in fact within an intron of the gene, downstream by more than 570 kb from the TSS. Capture Hi-C data from the same tissue in which the aiQTL was supported (aorta) suggests a chromatin interaction linking the region containing the aiQTL with the promoter (S3 Fig). We fitted our models to all *cis*-eQTLs reported by the GTEx consortium for all GTEx tissues [13]. For 19% of the significant *cis*-eQTL-gene associations, the null hypothesis was rejected, providing support for a direct *cis*-acting effect of the eQTL on the gene. This proportion is much higher for stronger eQTLs and for eQTLs close to the eGene (as is evident from Fig 3). We note that, for some of the remainder, the eQTL is also likely to act in *cis* but with insufficient allele-specifically mapped reads in individuals heterozygous for the eQTL to positively confirm an aiQTL.

## Allelic imbalance across samples and tissues

In order to compare the extent of allelic imbalance between individuals and across tissues, we estimated a single value of the $\alpha$ parameter of the symmetric beta-binomial distribution per

sample. This allowed us to estimate the distribution of the binomial parameter for the sample across all genes. Samples with high values of $\alpha$ tend to have values of the binomial parameter that are close to 0.5 and, correspondingly, to have relatively balanced expression of the two alleles of their genes. Samples with lower values of $\alpha$ have a more diffuse distribution of the binomial parameter. This is consistent with a tendency to have imbalanced gene expression, with values of the binomial parameter far from 0.5 occurring with relatively high probability. We note that this analysis is sensitive to all sources of imbalance (including imprinting and random monoallelic expression) and is not restricted to imbalance with a genetic cause. Nonetheless, it is perhaps surprising that we found that overall heterozygosity (measured as the number of loci at which an individual was heterozygous) was not strongly correlated with sample-level allelic imbalance (unadjusted P >0.05 in all but three GTEx tissues). When we compared individuals by sex, we did not find striking differences in the extent of allelic imbalance, though the comparison survived multiple test correction for two esophageal tissues:

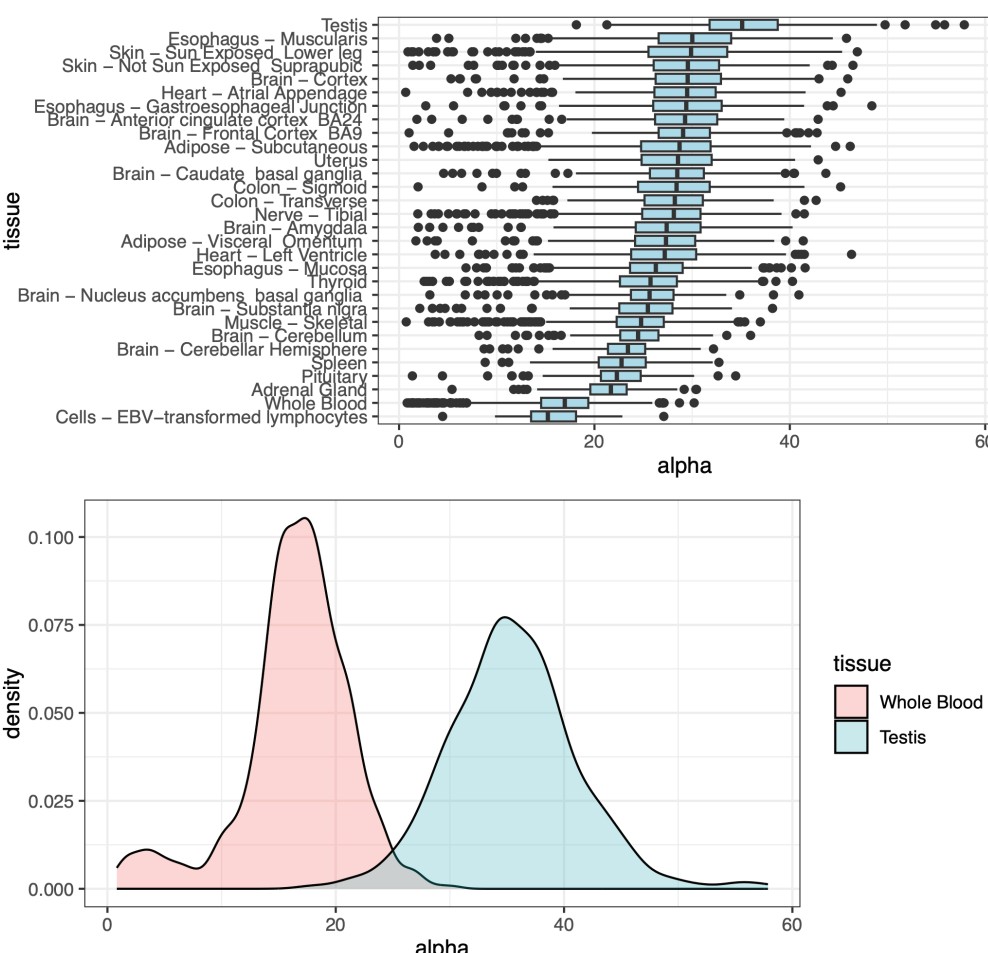

**Fig 4. Variation in the extent of allelic imbalance across samples and across tissues.** The results in this figure were obtained by fitting a single value of the $\alpha$ parameter of the symmetric beta distribution for each sample. A) Boxplots of the $\alpha$ parameter inferred for each sample across GTEx tissues. B) Density plots of the $\alpha$ parameter estimated for whole blood (red) and testis (blue) samples.

Esophagus - Muscularis and Esophagus - Gastroesophageal Junction (Holm-adjusted P value = 0.014 and 0.029, respectively).

We also tested for an association between allelic imbalance and age. We hypothesized that the accumulation of damage to DNA and/or somatic mutations could contribute to increased imbalance in gene expression between alleles with time; this would be reflected in a negative correlation between age and the fitted value of the $\alpha$ parameter of the symmetric beta-binomial distribution for the samples. Indeed we found weak negative correlations that survived multiple testing correction (using the Holm method) for three tissues: Uterus, Sigmoid colon and Esophagus muscularis (adjusted P = 0.0010, 0.0051, 0.016, respectively; S4 Fig). For one tissue, whole blood, the correlation was significant in the opposite direction (adjusted P = 0.0014). This may be due to changes in the cellular composition of blood over time (see below). There were substantial differences in the extent of imbalance observed across tissues (Fig 4), with the lowest values of $\alpha$ (corresponding to the greatest amount of allelic imbalance) found in participant-derived cell lines, followed by whole blood, and the highest value of $\alpha$ (least imbalance) observed in testis. Given the genomic and transcriptomic alterations that are found in cell lines relative to untransformed cells [37], the elevated imbalance in cell lines was not surprising. Indeed, it has been noted [38,39] that the high degree of clonality of Lymphoblastoid cell lines (LCLs) means that random monoallelic expression can confound studies of ASE using LCLs. Both the elevated imbalance in whole blood and the reduced imbalance in testis may be explained by the proportions of immune cells in these tissues (see Discussion).

## Allelic imbalance correlates with inferred cellular composition in whole blood

We performed gene expression deconvolution on the whole blood gene expression data using CIBERSORTx [28] and tested for correlations between the $\alpha$ parameter of the symmetric beta-binomial distribution across samples and the inferred proportion of each cell type. Several of the constituent cell types were highly significantly correlated with the $\alpha$ parameter, suggesting differences in the extent of allelic imbalance between cell types (Table 1). The strongest positive correlation (corresponding to reduced allelic imbalance) was with CD8 T

**Table 1. Spearman correlation between allelic imbalance and inferred blood cell type proportions.**

| Cell type | Spearman $\rho$ | P-value[1] |
|---|---|---|
| CD8 T cells | 0.41 | $3.0 \times 10^{-29}$ |
| Naive CD4 T cells | -0.31 | $1.2 \times 10^{-16}$ |
| Resting memory CD4 T cells | 0.29 | $2.3 \times 10^{-14}$ |
| Follicular helper T cells | 0.18 | $1.4 \times 10^{-6}$ |
| Regulatory T cell | 0.22 | $5.6 \times 10^{-9}$ |
| Gamma-delta T | -0.28 | $3.6 \times 10^{-13}$ |
| Resting NK cells | 0.33 | $1.9 \times 10^{-18}$ |
| Activated NK cells | 0.36 | $2.6 \times 10^{-22}$ |
| Monocytes | 0.20 | $1.2 \times 10^{-7}$ |
| M2 macrophages | -0.24 | $6.6 \times 10^{-10}$ |
| Resting mast cells | -0.32 | $3.8 \times 10^{-17}$ |
| Activated mast cells | 0.36 | $5.8 \times 10^{-22}$ |
| Eosinophils | 0.31 | $3.3 \times 10^{-16}$ |
| Neutrophils | -0.44 | 0 |

[1] Only cell types significantly correlated (P < 0.05) with allelic imbalance are shown (from a total of 22 blood cell types analyzed)

cells and the strongest negative correlation (corresponding to increased allelic imbalance) was with neutrophils (Table 1). The inferred proportions of each cell type were also correlated with one another (S5 Fig), complicating the task of attributing differences in allelic imbalance to a specific cell type. When we included the inferred proportion of neutrophils or CD8 T cells in a linear model, the relationship between allelic imbalance and age (reported above) was no longer significant, suggesting that the decreased allelic imbalance with age reported above may, indeed, be a result of changes in cell type proportions with age.

## Discussion

Genetic variants that affect gene expression (eQTLs) make a substantial contribution to phenotypic variability and complex disease susceptibility [8]. Such variants can act in *cis* or in *trans*, but more attention has been given to *cis*-acting variants because detecting *trans*-acting variants and distinguishing them from indirect effects is more challenging. Variants that are co-located with the genes with which they are associated are often assumed to act in *cis*; however, there is no guarantee that a variant collocated with its target gene (referred to here as a proximal eQTL) acts in *cis* and it is not clear what the precise relationship is between the physical distance of an eQTL from its target gene and the probability that it acts in *cis*. More generally, although examples of enhancers that regulate genes over very large distances have been reported (more than a megabase in some cases [35,36]), it is not clear how common this is. Here, we propose a statistical model to assess whether the extent of allelic imbalance at a target gene is associated with the genotype of an eQTL and refer to a locus that is associated with the extent of imbalance as an allelic-imbalance QTL (aiQTL). This model can be used to assess the evidence that a proximal eQTL acts in *cis*.

Our model uses a mixture of symmetric beta-binomial distributions, which facilitates statistical tests based on differences in the mixing proportions between sample groups, defined by the genotype of the putative aiQTL. We also explored the use of a model that fits independent beta-binomial distributions to the genotypic groups (see Data and Methods); however, such a model is sensitive to individual samples with very large numbers of allele-specifically mapped reads and can give misleading results (e.g. suggesting an aiQTL when a single sample in one of the genotypic groups has high imbalance, resulting from a cause unrelated to the genotypic groups). Tests based on the weights in the mixture model avoid this problem and appropriately weight the evidence for a tendency for more or less imbalance in one of the groups, compared to the other. Restricting to symmetric beta-binomial distributions has two important advantages for our model: it facilitates the detection of aiQTLs that are far away from their target gene, because there is no assumption of linkage disequilibrium between the aiQTL and target gene and no requirement of phased data; in addition, the symmetric beta-binomial distribution has just a single free parameter, reducing the challenge of fitting the models and increasing statistical power, compared to fitting a mixture of general beta-binomial distributions.

Our aiQTL model may also facilitate exploration of the proportion of the *cis*-regulatory effects that operate through a particular genetic variant. For example, in the case of some aiQTLs, the fitted mixture model has one very high value of the $\alpha$ parameter, consistent with a subset of samples with almost no allelic imbalance (binomial parameter very close to 0.5) and a much lower value of $\alpha$, corresponding to samples with high imbalance. We identified examples of eQTLs for which the estimated mixture of the low $\alpha$ component is close to one for the heterozygotes, while the estimated weight of the high $\alpha$ component is close to one for homozygotes. This is consistent with the majority of the *cis*-regulatory effects acting on this gene involving the aiQTL (or variants that are in close linkage disequilibrium with it). In other

cases, the imbalance is greater for heterozygotes (lower $\alpha$), but the value of $\alpha$ was also relatively low in homozygotes, suggesting substantial imbalance remains after accounting for the effect of the aiQTL (and other variants that may be in LD with it). This approach to studying the contribution of a genetic variant to allelic imbalance provides an alternative to methods that estimate allelic fold change from estimated eQTL effects [14,40].

An earlier method to assess the relationship between an eQTL and ASE, proposed by Zou *et al.* [16], defined aiSNPs as SNPs for which the heterozygous genotype is associated with allelic imbalance of a nearby gene. An important distinction between this method and ours is that Zou *et al.* treat allelic imbalance as a binary quantity (a gene in a given individual either does or does not display allelic imbalance, depending on the results of a statistical test applied to the allele-specifically mapped sequence reads in that individual). By contrast, our method does not require a binary determination of whether a gene is imbalanced in a given individual. A key advantage of not making this binary determination is that our method can be effective even for lowly expressed genes, as it can accumulate evidence across multiple individuals, even when there is insufficient evidence of allelic imbalance in any given individual, due to a low total number of sequence reads obtained from that individual. There is also a bias in the use of a binary classification of samples as exhibiting allelic imbalance or not, in that there will be more power to detect imbalance in individuals with higher expression of a gene. This can result in a *trans*-eQTL being inferred to act in *cis* (S1 Fig). In addition, a method that first performs a test of ASE and then compares proportions of samples displaying significant ASE between groups defined by the genotype of an eQTL may be unable to detect an aiQTL if there are other contributors (genetic or non-genetic) to ASE. In such a case most or even all samples could exhibit statistically significant ASE, particularly for more highly expressed genes for which the higher read counts provide greater power to reject a null hypothesis of balanced expression of the two alleles. A statistical test designed to detect differences in the proportion of samples displaying ASE may not be sensitive to a difference in the extent of imbalance in heterozygotes of the aiQTL compared to homozygotes. Lastly, our method has the advantage that it is amenable to studying variation in the prevalence of allelic imbalance across individuals and across tissues, as the estimate of the $\alpha$ parameter of the symmetric beta distribution should not be affected by differences in the number of mapped reads between samples.

Fitting symmetric beta-binomial distributions to individual samples, also enabled us to investigate factors that may influence the extent of allelic imbalance in a sample. The most obvious factor that could influence the extent of imbalance in a sample is the extent of heterozygosity. The occurrence of genetic variation in regulatory regions could result in an increased tendency towards expression imbalance in individuals with a higher degree of heterozygosity; however, we did not find strong evidence for this. Across all tissues, the greatest allelic imbalance was found in whole blood with the least imbalance in testis. We hypothesize that both of these results may reflect the fact that several immune-related genes show a high degree of random mono-allelic expression [41]. It is important to note that the individual-level analysis does not detect aiQTLs, but instead is sensitive to the degree of imbalance across genes within a single sample. Random mono-allelic expression will contribute to this (though not to aiQTL signals, which require differences in the extent of imbalance between genotypic groups). This may cause samples of tissues with a high proportion of immune cells (such as whole blood) to show a greater tendency for allelic imbalance and tissues with a lower rate of immune activity (such as testis, which is immune privileged [42]) to have less allelic imbalance, on average.

At the individual sample level, there was limited evidence of differences between the sexes and correlations with age. We considered that correlations with age that were observed may

reflect cell type composition. For example, changes in the abundance of different immune cell types with age could result in changes in the degree of imbalance in a sample. To explore this hypothesis for the case of blood, we performed gene expression deconvolution and tested for associations between the extent of imbalance (via the $\alpha$ parameter of our model) and the inferred abundance of the different blood cell types. The correlation with age for blood disappeared altogether when we took cell type proportions into account. The absence of a general tendency for an association between allelic imbalance and age suggests that somatic mutations and the accumulation of damage to DNA over time do not cause an increase in expression imbalance between alleles. However, the correlations between cell type proportions and the degree of imbalance could provide useful insights into differences between cell types in the extent of allelic imbalance. The correlations observed in the blood are consistent with a greater degree of imbalance on average across genes in neutrophils, for example, and a lower degree of imbalance in CD8 T cells. Some differences were observed between the same cell type in different states; for example, the results are consistent with greater imbalance in gene expression in naïve CD4 T cells and lower imbalance in resting memory CD4 cells. These results could simply reflect differences in selective constraints acting on gene expression between the genes that are up- or down-regulated between these cell states, such that the allelic imbalance contributed by *cis*-acting variants is greater or less in one cell type or cell state. It could be of interest to identify the genes and gene sets that show increased or reduced imbalance between cell states to explore potential non-genetic sources of imbalance that are specific to a cell type, such as differences in random monoallelic expression.

The identification of aiQTLs has the capacity to complement existing methods that are used to study genetic variation in gene expression. Several previous methods to infer eQTLs combine evidence from variation in gene expression across samples with variation in expression level between alleles [17,18,23]. We propose that there is a value to complementing this approach with a separate analysis of the genetics of imbalance itself, treated as a quantitative trait. The aiQTL signal is orthogonal to the evidence of an eQTL from variation in expression level across samples and complements expression level analysis by providing the capacity to test explicitly whether the eQTL effect arises in *cis*. We could make use of only a small fraction of mapped RNA-Seq reads to infer aiQTLs, because our analysis is restricted to reads that can be mapped to a specific allele, which requires reads that span heterozygous variants. Downsampling the data suggests that additional samples would reveal further aiQTLs (S6 Fig) and this is consistent with the fact that we could detect an aiQTL effect for only around half of the strong eQTLs located close to the TSS (Fig 3). Given developments in long-read sequencing technologies, future transcriptomic datasets are likely to have much greater power to detect aiQTLs, because a much larger proportion of reads will span heterozygous loci. This will enhance the power to explore *cis*-regulatory effects by inferring aiQTLs.

## Supporting information

**S1 Fig. Comparison with other methods based on simulation results.** A) Comparison of the power of our method (referred to in the legend as 'model') and three other methods that can test for an association between ASE and the genotype of an eQTL (thereby inferring that the eQTL acts in *cis*). The results shown are based on the simulations illustrated in Fig 2A,B and correspond to an aiQTL of moderate effect size. B) False positive inferences of an aiQTL as a function of the size parameter of the negative binomial distribution. These results are based on simulations that are equivalent to those shown in Fig 1D, but with a *trans*-acting

eQTL (resulting in an increase in gene expression by a factor of 1.3). The red dashed line shows the significance level of the tests (i.e. the expected false positive rate).
(PNG)

**S2 Fig. Disrupt plot showing the comparison of the number of aiQTLs detected by four different methods, including the one developed here (referred to as 'model' in the plot).** The strongest eQTL, with minor allele frequency of at least 0.05, was tested for each expressed gene in Whole Blood. Only cases where the eQTL could be tested using all four methods were considered.
(PNG)

**S3 Fig. Long-range aiQTL.** An example of a long-range aiQTL supported by a promoter Capture Hi-C data in the same tissue (aorta), generated using the 3D genome server (3dgenome.fsm.northwestern.edu/chic.php). The affected gene is *GPC6*, a glypican gene on chromosome 13 that is well over a megabase in length. The red arc indicates a chromatin interaction derived from the Capture Hi-C data that links the promoter region to the aiQTL, located in intron two of the gene and separated from the promoter by more than 570 kb. Coordinates shown are for the hg19 human genome assembly.
(PNG)

**S4 Fig. Scatterplots illustrating the relationship between the $\alpha$ statistic and subject age in four GTEx tissues.** In the case of (A) Uterus, (B) Colon - sigmoid and (C) Esophagus - muscularis the $\alpha$ statistic showed weak negative correlated with age, consistent with a tendency towards increased allelic imbalance in older subjects. In Whole blood (D), the correlation was in the opposite direction. P values show within each of the panels have been adjusted for multiple testing using the Holm method.
(PNG)

**S5 Fig. A heatmap illustrating the Pearson correlation between the inferred proportions of distinct immune cell types across whole blood samples from GTEx.**
(PNG)

**S6 Fig. Downsampling applied to aiQTLs detected in Whole Blood (nominal P < 0.01).** A) $\log_{10}$ P-value of the aiQTL as a function of the downsampling sample size. Lines are coloured by the $\log_{10}$ P-value obtained using all 670 samples. The dashed line corresponds to the P-value of 0.01. The y-axis is truncated to a minimum $\log_{10}$ P-value of -30. B) The proportion of the aiQTLs that were detected, as a function of the downsampling sample size. Error bars show twice the standard error of the proportion (corresponding approximately to a 95% confidence interval).
(PNG)

## Acknowledgments

We are grateful to Liam Seoighe for assistance with manuscript revision.

## Author contributions

**Conceptualization:** Cathal Seoighe.

**Data curation:** Cathal Seoighe.

**Formal analysis:** Cathal Seoighe, Mehak Chopra.

**Funding acquisition:** Cathal Seoighe.

**Investigation:** Cathal Seoighe, Seán Connaire.

**Methodology:** Cathal Seoighe.

**Project administration:** Cathal Seoighe.

**Resources:** Cathal Seoighe.

**Software:** Cathal Seoighe.

**Supervision:** Cathal Seoighe.

**Validation:** Cathal Seoighe.

**Visualization:** Cathal Seoighe.

**Writing – original draft:** Cathal Seoighe.

**Writing – review & editing:** Cathal Seoighe.

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
