## [Decision Letter · Decision Letter 0]

15 Nov 2024

PGENETICS-D-24-01127

Probing the limits of cis-acting gene regulation using a model of allelic imbalance quantitative trait loci

PLOS Genetics

Dear Dr. Seoighe,

Thank you for submitting your manuscript to PLOS Genetics. As with all papers, your manuscript was reviewed by members of the editorial board. Based on our assessment, we have decided that the work does not meet our criteria for publication and will therefore be rejected. If external reviews were secured, reviewers' comments will be included at the bottom of this email.

We are sorry that we cannot be more positive on this occasion. We very much appreciate your wish to present your work in one of PLOS's Open Access publications. Thank you for your support, and we hope that you will consider PLOS Genetics for other submissions in the future.

Yours sincerely,

James J Cai

Guest Editor

PLOS Genetics

Paula Cohen

Section Editor

PLOS Genetics

Aimée Dudley

Editor-in-Chief

PLOS Genetics

Anne Goriely

Editor-in-Chief

PLOS Genetics

**Additional Editor Comments (if provided):**

**Reviewers' Comments (if peer reviewed):**

Reviewer's Responses to Questions

**Comments to the Authors:**

Reviewer #1: In this manuscript, Seoighe et al. introduce a new computational method for identifying cis-acting genetic variation and apply it to GTEx data. The work is partially motivated by the important distinction between variants that are cis-acting versus simply located close to the gene. The method performs well in simulations, and its application to real data revealed that local eQTLs at a distance of more than 500 kb from the TSS of the affected gene are unlikely to act in cis – an interesting observation. Application to different tissues and estimated immune cell components are also good additions to the paper.

Overall, this is a strong, focused paper that does a nice job introducing a method that is a useful addition to the field. The paper is well written and the figures are clear. I do not have many comments.

My only somewhat major comment is that I wasn’t able to fully follow the description of the mixture model. Specifically, what are the two components? They appear not to be the two specific A and B alleles, since those are assigned arbitrarily in different samples. They also do not appear to be the homozygous and heterozygous samples since those are later compared using two estimates of π. So what are they? What do alpha1 and alpha2 refer to? A clarifying sentence or two should take care of this.

Further, on page 9: “examples of enhances that act over large distances” should be “enhancers”.

Reviewer #2: Seoighe et al. have developed a new statistical method for analyzing allelic imbalance in gene expression and particularly genetic variants that contribute to this (aiQTLs). The method seems to provide some improvements over analogous prior methods, although systematic benchmarking is not provided. In addition to simulations, they apply the method to GTEx data, and describe how allelic imbalance changes as a function of variant distance to TSS, as well as overall as a function of age and cell type composition.

While the paper has merits – the method seems fine and the analysis indicates some interesting findings – I find the presentation to be quite shallow. For a methods paper I’d need to see better benchmarking, stronger simulations, and overall more work to understand when and ho the method works and what its limitations are. As is, it’s difficult to be estimate whether it actually represents a step over state of the art. The analysis of GTEx data is fine, but the presentation and novelty of the results is somewhat lacking. I’m not even holding the bar at finding some truly novel phenomena, but not all the conclusions are well supported by the data provided, and/or additional steps could have bene taken for deeper insights.

I also have some specific comments, in no particular order of importance:

- coloc-eQTLs / colloc-eQTL (both spellings occur) – I would suggest another name.

- Page 2 intro on using ASE to identify rare variants – some references might be in order here?

- “linkage disequilibrium between the eQTL and affected gene” – you probably mean “between the eQTL and variants in the affected gene”

- “our results suggest that the majority of eQTLs between half a megabase and one megabase from the eGene do not act in cis.” This is a strong and extraordinary claim, contrary to biological expectations. It is not sufficiently supported by the data provided. Rather, I think that the results presented are fully compatible with the known weaker cis-regulatory effects of variants at larger distances from TSS (due to weaker but real enhancer effects), which leads to lower power to detect them as aiQTLs.

- The simulations assume full lack of LD between the aiQTL and the transcribed variant. Could the authors simulate situations with partial LD, since this is often going to be the case? This may lead to loss of power when not all genotype classes are observed at randomly expected frequencies

- The fold change effect sizes are quite strong (1.2) even for the weakest simulated effects. It would be good to have more simulations and/or e.g. downsampling of empirical data to test how properties of the data affect the performance of the method.

- It’ not true that the extent of allelic imbalance for cis-eQTLs as the function of distance from TSS hasn’t been tested before; see GTEx main paper 2017. The results are somewhat corroborative (although see next point), but this should be referenced and described

- I don’t think I quite understand the model or the data behind Figure 2b. Writing down the actual equation would be helpful. It seems to me that samples with just 2 allelic reads are included but at that level the data is extremely sparse. I’m not sure I understand this right, but if from this logistic model one derives coefficients to then plot the probability of Fig 2a for the given parameters, then the result shown there is extremely specific to this set of parameters and not really biologically interpretable across the parameter space. If the y-axis is to be taken with biological interpretation of “suggest that direct cis-acting effects at a distance of 1Mbp up or downstream of the TSS are rare”, I note that the probability is only 0.5 even at TSS – and it would be a highly implausible claim that half of eQTLs at TSS don’t act in cis.

- The point above raises the question: if/since allelic analysis is often underpowered to find evidence of even a true cis-effect, isn’t it the case that this analysis can support an eQTL being in cis, but absence of evidence doesn’t mean evidence of absence? If this is the case, can we actually use the approach described here for the purpose of determining whether an eQTL acts in cis or not?

- The LM22 leukocyte reference might not be that great for whole blood deconvolution, as the GTEx whole blood contains other cell types than leukocytes; particularly hemoglobin expression is very high.

- Note that the GTEx LCLs have a high degree of clonality (see e.g. Baral et al. 2015 Genome Research) which will lead to allelic imbalance from random monoallelic expression. This explains the result for the LCLs in Fig 3a.

- Isn’t clonal hematopoiesis a potential reason for reduced allelic imbalance in blood, associated with age and cell type composition?

- I have a feeling that more could be done on the cell type specificity of random monoallelic expression that is cites as the likely main cause of the differences in allelic imbalance. The references to prior literature seem shallow. Isn’t there accessible data (RNA-seq from sorted blood cell populations) that would allow testing some of the hypotheses that are posed?

- The section “Allelic imbalance across samples and tissues” describes a lot of results that are not shown in any display item. It would be interesting to see these, as main figure panels or in the supplement.

- In general, the figure panels should include plotted color legends rather than having these described in the legend text; the current style is very difficult to decipher. It’s also extremely confusing when the meaning of colors switches between panels, e.g. in Fig 1 red is sometimes weak, sometimes strong effect.

- Fig 1a: I don’t understand the x-axis label (and consequently, the entirely figure).

- Fig 1b: Do I understand correctly that strong ASE is linked to lower power? How, why?

- “Genetic drift in regulatory regions could result in an increased tendency towards expression imbalance in individuals with a higher degree of heterozygosity” I fail to see how genetic drift (in the exact definition of the phrase) has a role here. Functional genetic variation can arise from a number of population genetic processes, and nothing in the analyses here addresses these.

Reviewer #3: In this paper, the authors propose a statistical model for identifying aiQTLs to investigate the limits of cis-acting gene regulation. Their model, based on a symmetric beta-binomial distribution, is designed to detect cis-regulatory eQTLs without relying on linkage disequilibrium, making it particularly effective for identifying long-range regulatory effects. Their methods are validated using both simulated data and real data from the GTEx consortium. The advantages of these methods include their robustness against false positives, improved power to detect cis-acting eQTLs even at large genomic distances, and adaptability for tissues with varying gene expression levels. Overall, this approach can be a very valuable tool for mapping regulatory variants across the genome. The manuscript is well-organized, although the clarity of the presentation could be significantly improved.

1. The "Model and Implementation" section is too brief and lacks technical details. For example, descriptions of the data (not individual datasets, but the form and setting of the data) and overall settings of the problem are absent. There should be an explicit description of the number of samples, the A allele, etc. Ideally, a diagram or graph should be included to illustrate the form of the data. This diagram should display the data, the problem to be solved, and the differences in assumptions made by their methods compared to previous methods. Additionally, the authors need to provide the log-likelihood function of the model, which was used to estimate the parameters, and clearly mathmatically formulate the null and alternative hypotheses.

2. In the same section, the explanation of why setting alpha = beta is unclear, although briefly mentioned. When alpha = beta, the beta distribution always has a mean of 1/2; what implications does this have (for the symmetry of the beta-binomial distribution?)? Is this assumption necessary to align with biological facts? If such an assumption arises purely to control the number of parameters used, could it compromise the biological interpretation or even complict with the biological fact?

3. It is commendable that the authors simulated data that do not exactly follow the assumed distribution, which greatly aids in evaluating the robustness of the proposed model. However, in the simulated data, the authors did not compare the proposed method with existing methods or other regression/mixture models, making it difficult to determine whether the achieved power is high.

4. Similarly, in real data analyses, a detailed comparison with existing methods and an explanation of why their methods are superior are needed. The comparison should include a Venn diagram-like illustration of the agreement/difference in findings from different methods, as well as a detailed analysis of those differences--which findings make more sense?

**Have all data underlying the figures and results presented in the manuscript been provided?**

Reviewer #1: Yes

Reviewer #2: Yes

Reviewer #3: None

PLOS authors have the option to publish the peer review history of their article (what does this mean?). If published, this will include your full peer review and any attached files.

Reviewer #1: No

Reviewer #2: No

Reviewer #3: No

---

## [Decision Letter · Decision Letter 1]

23 Feb 2025

PGENETICS-D-24-01127R1

Probing the limits of cis-acting gene regulation using a model of allelic imbalance quantitative trait loci

PLOS Genetics

Dear Dr. Seoighe,

Thank you for submitting your manuscript to PLOS Genetics. After careful consideration, we feel that it has merit but does not fully meet PLOS Genetics's publication criteria as it currently stands. Therefore, we invite you to submit a revised version of the manuscript that addresses the points raised during the review process.

Please submit your revised manuscript within 30 days . If you will need more time than this to complete your revisions, please reply to this message or contact the journal office at plosgenetics@plos.org. Please include the following items when submitting your revised manuscript:

We look forward to receiving your revised manuscript.

Kind regards,

James J Cai

Guest Editor

PLOS Genetics

Paula Cohen

Section Editor

PLOS Genetics

Aimée Dudley

Editor-in-Chief

PLOS Genetics

Anne Goriely

Editor-in-Chief

PLOS Genetics

**Additional Editor Comments :**

Thanks for your effort in revising your manuscript. Please address the remaining comments raised by Reviewer #2, including the suggested changes regarding "The different meaning of figure color...". Your manuscript should then be ready for publication upon the editorial review of your responses to those comments.

**Reviewers' comments:**

Reviewer's Responses to Questions

Reviewer #1: Thanks to the authors for addressing all my comments.

Reviewer #2: Seoighe et al. have revised the manuscript and it has improved a lot. They provide important further clarity and confidence in the method that seems to perform quite nicely. I have relatively minor comments related to the biological results:

I will number my response by the numbers given my the authors in their response:

# 5, #9 It’s still a little unclear to me whether the authors claim that absence of evidence of an eQTL acting in cis is evidence of absence of a cis effect (as a function of distance from TSS). This is the main concern of the biological results that I raised, and the response and revision is not quite clear on this. As far as I know, regulatory variants function either in cis or trans, and thus claiming that X% has been shown not to act in cis is a claim that X% has been shown to act in trans (sometimes the manuscript states this too), and I do not think the data justifies this claim. While the authors include some covariates in the analysis of aiQTL as a function of TSS distance to control for differences in power, I am not convinced that these fully capture the differences of eQTLs close to TSS and further at CREs. For the latter, we expect smaller effect sizes, which eQTL p-value only partially captures (for that I’d want to see effect size, preferably as allelic fold change, included in the model). Furthermore, enhancer CREs and eQTLs are more context-dependent, so we expect to see more variability especially in blood that has a particularly high cell type heterogeneity between individuals. LD patterns will also differ. All these factors are very difficult to control to their full extent to allow making strong quantitative claims. Altogether, while I believe that the decay of cis-effects as a function of distance is a real phenomenon, a part (un unknown magnitude) of the observed pattern is likely to be explained by differences in power. This has to be consistently and transparently stated. I strongly urgent the authors to revise statements like “our results suggest that the majority of eQTLs at distances more than 500 kb from the TSS of the target gene are likely to act in trans” (from the abstract) to something like “support for cis-effects cannot be detected for the majority of eQTLs at distances more than 500 kb from the TSS, indicating that many of these eQTLs are likely to act in trans“.

#12: Apologies for a typo in my first review; I referred to Baran and not Baral Genome Research 2015 (PMID: 25953952). The Plagnol reference is fine, but Baran et al. shows this specifically for GTEx LCLs; the extent of clonality varies between data sets and is usually high in GTEx.

#16: The different meaning of figure colors as the reason not to have them in figure inserts is a strange logic to me – having them in figures is clearer and faster to read, and if the meaning changes between panels, then you need the legend even more (and/or selecting different colors). It’s very cumbersome to comb through legend text to find out the meaning of colors. But I’ll leave this to the editor.

“However, the relationship between the distance separating a proximal eQTL from its eGene and the likelihood that the eQTL acts in cis has yet to be determined” – this is not true in the light of data in ref 12, is it? This is easy to rewrite to “underexplored” which is certainly true.

“The figure also suggests a greater probability of variants downstream of than upstream acting in cis over long distances (this could be because, for some long-range interactions, distance from the gene may be more relevant than distance to the TSS).” I’d hypothesize that this is due to some variants in the transcribed region have post-transcriptional effects e.g. on RNA stability leading to eQTLs/aiQTLs, as opposed to distal upstream effects driven solely by enhancers.

Reviewer #3: The authors have successfully addressed all my previous concerns.

**Have all data underlying the figures and results presented in the manuscript been provided?**

Reviewer #1: None

Reviewer #2: Yes

Reviewer #3: None

PLOS authors have the option to publish the peer review history of their article (what does this mean?). If published, this will include your full peer review and any attached files.

Reviewer #1: No

Reviewer #2: No

Reviewer #3: No

**Figure resubmission:**
---

## [Editor Report · Decision Letter 2]

27 Mar 2025

Dear Dr Seoighe,

We are pleased to inform you that your manuscript entitled "Probing the limits of cis-acting gene regulation using a model of allelic imbalance quantitative trait loci" has been editorially accepted for publication in PLOS Genetics. Congratulations!

Yours sincerely,

James J Cai

Guest Editor

PLOS Genetics

Paula Cohen

Section Editor

PLOS Genetics

Aimée Dudley

Editor-in-Chief

PLOS Genetics

Anne Goriely

Editor-in-Chief

PLOS Genetics

Comments from the reviewers (if applicable):

**Data Deposition**

http://datadryad.org/submit?journalID=pgenetics&manu=PGENETICS-D-24-01127R2

**Press Queries**

---

## [Editor Report · Acceptance letter]

PGENETICS-D-24-01127R2

Probing the limits of cis-acting gene regulation using a model of allelic imbalance quantitative trait loci

Dear Dr Seoighe,

We are pleased to inform you that your manuscript entitled "Probing the limits of cis-acting gene regulation using a model of allelic imbalance quantitative trait loci" has been formally accepted for publication in PLOS Genetics! Your manuscript is now with our production department and you will be notified of the publication date in due course.

With kind regards,

Anita Estes

PLOS Genetics

On behalf of:
